# "I feel good… I knew that I would…": The role of self in musical reward across cultures

Jonathan Tang *

Department of Music, University of Sheffield, Sheffield, United Kingdom

* tang.jon@gmail.com

## Abstract

Listening to music can be a rewarding experience for many. Research has shown that multiple factors influence musical reward including personality, age, and musical expertise. However, the role of culture in shaping musical reward remains under-explored. Most cross-cultural studies in music psychology have compared individuals from different countries. This study adopted a novel approach by examining self-construal, an individual-level explanation for cultural differences, in relation to musical rewards associated with favourite music across cultures. A cross-sectional online questionnaire was administered to 435 participants. Results from the multilevel regression analyses, using the two-dimensional model of self-construal, revealed that only within-region variation of interdependent and independent self-construals, not between-region variation of interdependence and independence, were positively associated with musical reward. Specifically, both self-construals were associated with emotion evocation and social reward, while independent self-construal was associated with musical seeking, mood regulation, and sensory-motor subtypes. When applying the eight-dimensional model of self-construal, distinct self-construal profiles emerged in relation to different musical reward subtypes, with the interdependent pole of *connectedness to others* positively associated with most subtypes except for emotion evocation reward. These findings provide preliminary evidence that self-construal influences the types of rewards experienced across cultures. In particular, one's sense of self, whether construed as interdependent or independent, shapes the types of rewards experienced with favourite music. This study underscores the importance of incorporating specific cultural factors in cross-cultural research on musical reward. By examining self-construal, this work contributes to a more nuanced understanding of cultural diversity in music psychology.

## Introduction

As James Brown belts out, feeling good is one of the primary pleasures of music. Indeed, listening to music can be a pleasurable experience for many [1], but do

**Data availability statement:** The data underlying the results presented in the study are available in the University of Sheffield data repository, ORDA, at https://doi.org/10.15131/shef.data.28677104.

**Funding:** This work was supported by the Arts & Humanities Research Council (grant number AH/R012733/1) through the White Rose College of the Arts & Humanities.

**Competing interests:** I declare that no competing interests exist.

people from all cultures experience the same pleasure? Neuroimaging studies have found that both the dopaminergic reward system and emotional brain circuitry are activated when listening to highly pleasurable music [2]. These psychophysiological mechanisms could suggest that rewards derived from music listening are universal. Neuroimaging studies, however, tend to treat reward as a unitary phenomenon driven by biological substrates and disregard the multifaceted nature of musical experience involving the interaction of biological, psychological, and cultural factors [3]. Although music is universal [4], it is also an activity laden with cultural meanings, practices, and values [5,6]. It is therefore important to acknowledge cultural diversity in music cognition by examining how cultural factors influence musical reward.

In research on the psychology of music, factors such as nationality, country of residence, and ethnicity have been commonly used as proxies for underlying cultural differences [7–9]. However, the world today is increasingly interconnected and cultural contexts are melded together as individuals move from country to country [10]. Furthermore, culture does not solely exist in the countries where people live, but also in the ways they perceive, comprehend, and interpret their self (i.e., self-construal) [9,11–13]. Empirical evidence supports the theory that self-construal affects many aspects of behaviour including cognition and emotion [14]. Specifically, research in cultural neuroscience has shown that the same brain regions process both self-construal and reward [15,16]. Thus, cultural differences can be explained at the individual level through self-construal, potentially shedding light on the ways in which culture influences the pleasure to be gained from listening to music. Consequently, the aim of this study was to investigate the role of self in musical reward across cultures.

## Cultural determinants of musical reward

Scholars have different perspectives on reward depending on their cultural outlook [3]. Broadly speaking, musical reward can be defined as the pleasurable experience associated with music (hedonia) and the sense of fulfillment it brings to one's life (eudaimonia). For this paper, I adopt the hedonic perspective on musical reward and understand it to be a hierarchical concept in which different subtypes are subsumed under a higher-level unitary form of reward [17]. Mas-Herrero et al. delineated five subtypes of musical reward: *musical seeking*, pleasure gained from seeking information about music; *emotion evocation*, pleasure obtained from the feelings evoked by music; *mood regulation*, pleasure gained from using music to regulate affect; *sensory-motor*, pleasure derived from dancing or moving to music; and *social reward*, pleasure when bonding with others through music [18]. Cardona et al. added to this framework *absorption in music*, a pleasurable state of transcendence or complete immersion in music [19].

Research has shown that multiple factors influence musical reward and its subtypes, such as personality, age, gender, musical expertise, and music-cognitive traits [20]. Personality types, such as conscientiousness, agreeableness, neuroticism, and openness were significant independent factors that predicted musical reward as a whole [21]. Specifically, extraversion was positively associated with social reward and neuroticism was positively associated with mood regulation [22]. Other studies have

shown that age was negatively associated with various musical rewards, including the musical seeking, mood regulation, and sensory-motor subtypes [18,19]. Musical expertise has also been found to be positively associated with musical seeking and emotion evocation subtypes of musical reward. Taken together, the evidence suggests that musical reward is a differentiated phenomenon such that multiple intersecting factors influence the type of reward that is experienced when listening to music.

To my knowledge, no research to date has explicitly examined the influence of cultural factors on musical reward. Participants from different countries were recruited in the studies cited above: Spain and North America [18], India [22], China [21], and Germany [20]. Nevertheless, it would be premature to conclude that the determinants of musical reward and its subtypes are pancultural for two reasons. First, no cross-cultural comparisons were made even though participants came from different countries, so there is no evidence for or against the theory that musical rewards differ according to cultural context. Second, culture goes beyond country and nation state [10]. To propose valid hypotheses, it is imperative to articulate theoretical frameworks specifying the aspects of culture that are likely to influence musical reward [23].

## Culture, self, and reward

Culture is often associated with nation states. As such, researchers frequently operationalise culture using sociodemographic categories such as nationality, race, and ethnicity. Nationality refers to citizenship or permanent residency in a particular country, while race and ethnicity pertain to classification based on physical attributes and shared cultural heritage. I acknowledge that culture is not neatly bounded within these sociodemographic categories. Instead, culture is an untidy construct that includes an expansive set of material and symbolic concepts that give form and direction to behaviour [12]. Examples include cultural values, such as self-direction, benevolence, hedonism, and conformity [24]; cultural syndromes, such as tightness, active-passive, and honour [25]; and cultural mental programming, such as power distance, uncertainty avoidance, and long-term versus short-term orientation [26].

Several studies in music psychology have attributed cross-cultural differences in affective experiences of music to the norms and values inherent in collectivistic and individualistic societies [27–30]. For example, research on the uses and functions of music has shown that members of collectivistic societies (i.e., Kenya, Mexico, Philippines, Turkey, India, Hong Kong, Brazil, and Singapore) tend to use music for social purposes (e.g., diversion and social bonding) whereas members of individualistic societies (i.e., New Zealand, Germany, and the US) use music for self-centric purposes (e.g., emotion regulation and self-reflection) [31–33]. By using nationality and geographical boundaries to operationalise collectivism-individualism, these studies examined cross-cultural differences between cultural contexts in the form of nation states.

Other theories of culture encourage us to go beyond the cultural context and consider the role of self (or self-construal) in cross-cultural investigations [9]. Markus and Kitayama note that culture is not separate from the individual; it is a product of human activity [12]. Instead of comparing differences between groups according to the collectivist or individualist dichotomy, they explain cultural differences between individuals' perception, emotion, and behaviour according to their self-construal [34]. Cultural psychologists have theorised that there are two types of self-construal, and that these vary both within and between cultural contexts [12]. Individuals with an interdependent self-construal view themselves as socially embedded, such that their behaviour is contingent on their relationships with others. Individuals with an independent self-construal view themselves as unique and separate from others, meaning their behaviour is strongly driven by their own thoughts, feelings, and motivations. In other words, social harmony and interpersonal priorities are more important to individuals with interdependent selves, whereas personal preferences and intrapersonal priorities are more important for people with independent selves.

From this perspective, individuals are not seen as passive members of the cultures that they belong to, but as active agents consciously or unconsciously reflecting, reinforcing, and changing the cultures that they are part of [11–13]. In other words, people may possess self-construals that align with their cultural environment (e.g., an interdependent self-construal in a collectivistic context) as well as self-construals that contrast with it (e.g., an independent self-construal

in a collectivistic context). Previous studies in music psychology have investigated cross-cultural differences by comparing groups from various countries, which often assumes that individuals automatically adopt or adhere to the collectivistic or individualistic values of their broader cultural context. Measuring cultural differences at the individual level, by examining the self, offers a more nuanced approach to understand how culture influences musical reward because it is sensitive to variations between both individuals and groups, and specifies the exact psychological mechanisms through which culture shapes musical reward experiences.

Studies in cultural neuroscience have provided evidence that self-construal modulates brain activity during cognitive and affective processes [15,16]. For example, substantial overlap exists between the neural networks involved in processing self-construal and that of reward, suggesting that the perception of reward can be influenced by different types of self-construal [35]. Using a forced-choice gambling task, researchers have found that individuals with a dominant interdependent self-construal exhibit similar neural activation in the reward network in response to rewards for themselves and for friends and close others [36–38]. Conversely, individuals with a dominant independent self-construal show greater reward network activation in response to rewards for themselves than for others. These findings demonstrate that self-construal can influence whether rewards are experienced in response to pleasurable activities directed toward oneself or close others. However, because previous studies predominantly employed gambling tasks to examine reward experiences, it remains unclear whether self-construal similarly influences rewards derived from other pleasurable activities, such as listening to music.

## Aims of the study

The primary aim of the present study was to investigate the role of self in musical reward across cultures. Although researchers have identified several factors influencing musical reward, the significant role of culture remains underexplored. This study seeks to address this gap by examining how cultural factors influence musical reward in different cultural contexts. Furthermore, previous research has examined musical reward broadly without considering specific pieces of music. Yet individuals often report using particular pieces of music for specific activities [39,40], which may affect the type of musical reward they experience.

In this study, participants were asked to reflect on the types of rewards that they experience when listening to their all-time favourite piece of music. I acknowledge that favourite music is highly context-dependent and can change from one moment to the next [41]. Nonetheless, favourite music was chosen because of its unique significance to individuals, reflecting both individual and cultural influences from collectivistic and individualistic contexts. Studies have shown that music preferences are closely linked to personality, cultural identity, and cultural values [42–45]. For example, music preferences can signal meaningful information about racial identity; preferences for rap, hip-hop, and soul genres tend to be associated with Black individuals, whereas preferences for rock, alternative, pop, country, and folk genres are more commonly associated with White individuals [46,47]. Furthermore, personal and cultural values explained differences in music preferences more effectively than personality traits alone [48]. These findings suggest that an individual's prevailing self-construal may be reflected in their favourite music, which in turn shapes the musical rewards they gain from listening to it. This study addressed the research question: is self-construal, both between and within cultures, associated with different musical rewards when listening to favourite music?

## Theoretical framework and hypotheses

I situated this exploration within the meaning-making processes individuals engage in while listening to their favourite music. I theorise that music engagement is culturally inflected [9,49–52], as music and listeners exist in a symbiotic relationship within a particular context. Through ongoing, interactive exchanges between musical elements and listener perceptions, they continuously define and influence each other.

In this exploratory study, I adopted a post-positivist approach which strives towards objective knowledge while recognising that theories are limited, situated, and socially constructed [53]: musical reward [18] and self-construal [12]. Given the evidence that self-construal impacts reward experiences [35–38], I hypothesised that interdependent self-construal would be positively associated with social reward. In contrast, independent self-construal was hypothesised to be positively associated with musical seeking, emotion evocation, mood regulation, and sensory-motor rewards, as these subtypes appear to be more closely aligned with self-directed motivations and experiences [18,31–33].

I recognize that scholars have challenged the dichotomy of interdependent and independent selves as representative of cultures all around the world [54]. Alternative models have been proposed to capture the complexity of global variation of self across cultures [55–57]. Consequently, I drew upon an eight dimensional model of self-reported ways of being interdependent and independent [54,58], which include: *self-reliance vs. dependence on others, self-containment vs. connectedness to others, difference vs. similar to others, self-interest vs. commitment to others, consistency vs. variability, self-direction vs. reception to influence, self-expression vs. harmony,* and *decontextualised vs. contextualised self*. These dimensions represent distinct forms of interdependent and independent selves that result in unique self-construal profiles in different cultural contexts. Since empirical research has not yet incorporated alternative models of selfhood across cultures, this study adopted an exploratory approach using the eight-dimensional model as an extension and refinement of the two-dimensional model of self-construal.

## Method

### Participants

Participants were recruited in two ways. First, they were recruited through an online advertisement disseminated by the researcher and the researcher's contacts at universities in Singapore, China, Hong Kong, the UK, and the US, in which participants were informed that they could be entered into a draw to win one of five £10 (~US$13) Amazon gift cards. Second, they were recruited through Amazon Mechanical Turk (MTurk) administered by CloudResearch [59], and paid US$1 after completing the online questionnaire. Participants who did not complete the questionnaire or had missing responses were removed. Four hundred and thirty-five participants were included in the final analysis, of which participants reported 32 nationalities and 20 ethnicities, living in 11 different countries (see Table 1 and S1 Appendix). Ethical approval was received on 3 November 2022 and data collection took place between 27 November 2022 and 25 March 2023.

### Materials

The survey was administered in both English and Chinese (simplified and traditional) and all the measures used had been translated in previous studies. Musical reward was measured using the Barcelona Music Reward Questionnaire (BMRQ) [18]. The BMRQ consists of 20 items using a 5-point Likert-type scale from 1 (*strongly disagree*) to 5 (*strongly agree*) to measure five subtypes of musical reward: musical seeking, emotion evocation, mood regulation, sensory-motor, and social reward. Each subtype was measured using four items and a composite musical reward score was calculated by summing the five subtypes. The Chinese version of the BMRQ was taken from Wang et al. [21]. Cronbach's alpha for the present study (all participants combined) was.86.

Self-construal was measured using Singelis' Self-Construal Scale (S-SCS) [62] and Yang's Self-Construal Scale (Y-SCS) [58]. The S-SCS consists of 15 items to measure interdependent self-construal and 15 items to measure independent self-construal using a 7-point Likert scale from 1 (*strongly disagree*) to 7 (*strongly agree*). The Chinese version of the S-SCS was obtained from T. Singelis (personal communication, November 17, 2022) and had been used in previous studies [63,64]. Cronbach's alpha for the present study (all participants combined) was.84 and.82 for the interdependent and independent scales respectively.

**Table 1. Overall participant description.**

|  | *n* | *%* |
|---|---|---|
| Age (Years) | | |
| Mean (SD) | 35.76 (12.63) | |
| Gender | | |
| Transgender | 1 | 0.2 |
| Non-binary | 8 | 1.8 |
| Female | 212 | 48.7 |
| Male | 211 | 48.5 |
| Prefer not to say | 3 | 0.7 |
| Disability | | |
| Yes | 54 | 12.4 |
| No | 365 | 83.9 |
| Prefer not to say | 12 | 2.8 |
| Prefer to self-describe | 4 | 0.9 |
| Education Level | | |
| Primary or elementary school | 1 | 0.2 |
| Secondary or middle school | 8 | 1.8 |
| Higher secondary or high school | 89 | 20.5 |
| College or university | 223 | 51.3 |
| Postgraduate degree | 114 | 26.2 |
| Musical identity [a] | | |
| Non-musician | 87 | 20.0 |
| Music-loving non-musician | 159 | 36.6 |
| Amateur musician | 86 | 19.8 |
| Serious amateur musician | 43 | 9.9 |
| Semi-professional musician | 36 | 8.3 |
| Professional musician | 24 | 5.5 |
| Music Lessons (Years) | | |
| Mean (SD) | 4.26 (6.00) | |

[a]Musical identity was obtained using the Ollen Musical Sophistication Index (OMSI) [60] musician rank item. Musical identity was used because it is the single-item measure that best represents musical sophistication and musicality [61].

The Y-SCS consists of a total of 48 items using a 9-point Likert scale (from 1 = *doesn't describe me at all* to 5 = *describes me exactly*) with 0.5 as intervals (i.e., 1.5, 2.5, 3.5, 4.5) to measure eight different dimensions of self-construal: *self-reliance vs. dependence on others, self-containment vs. connectedness to others, difference vs. similar to others, self-interest vs. commitment to others, consistency vs. variability, self-direction vs. reception to influence, self-expression vs. harmony,* and *decontextualised vs. contextualised self*. Each dimension was measured using 6-items and a positive score reflected a tendency toward independence whereas a negative score reflected a tendency toward interdependence. As recommended by V. L. Vignoles (personal communication, August 8, 2022), each dimension's score was ipsatised (i.e., the score of each dimensions minus the overall average) to reduce the influence of acquiescent responding [65]. Cronbach's alpha for the present study (all participants combined) was.73 for *difference vs. similar to others*,.69 for *self-containment vs. connectedness to others*,.72 for *self-direction vs. reception to influence*,.70 for *self-expression vs. harmony*,.87 for *consistency vs. variability*,.66 for *decontextualised vs. contextualised self*,.80 for *self-reliance vs. dependence on others*, and.68 for *self-interest vs. commitment to*

*others*. The Chinese version of the Y-SCS was obtained from V. L. Vignoles (personal communication, November 15, 2022).

Personality was measured using the Big Five Inventory 10-item version (BFI-10) [66,67]. Personality was initially included to examine the unique contributions of self-construal to musical reward alongside other relevant factors. However, the BFI-10 demonstrated low internal consistency in the present sample, with Cronbach's alpha values ranging from.26 to.67. Consequently, this measure was excluded from subsequent analyses, as its inclusion could have undermined the robustness and reliability of the findings.

Musical expertise was measured using self-reported years of musical training and the musician rank item (i.e., *"Which title best describes you?"*) of the Ollen Musical Sophistication Index (OMSI) [60]. I used these as proxies of musical expertise because previous research found that they were the best single-item measures for estimating musical sophistication and musicality [61].

## Procedure

Prospective participants were invited to participate in the study through a link on the online advertisement or through MTurk. After participants clicked on the link, they read the participant information sheet and completed the written informed consent form before starting the online questionnaire. Participants were instructed to think about their all-time favourite piece of music first. They then completed the BMRQ with respect to that piece of music, the S-SCS, the Y-SCS, and demographic questions (e.g., age, gender, education level, and musical expertise). Participants also completed questionnaires for a different research project. The online questionnaire was hosted by Qualtrics$^{XM}$. Participation was voluntary and participants could skip any questions they did not want to answer. This study received ethical approval via the University of Sheffield's Ethics Review Procedure, as administered by the Department of Music (reference #049931).

## Data analysis

Participants were grouped by nationality, following arguments that national differences in collectivism-individualism shape self-construal [34]. Given the wide range of nationalities (see S1 Appendix), I applied the cultural distance hypothesis [68], which assumes that cultures are similar if they are close to one another, to cluster participants by geographic region. Subsequently, I conducted a multilevel regression analysis with participants nested within regions to assess whether S-SCS, both between and within cultures, was associated with different musical rewards. As an extension and refinement of the S-SCS, I conducted a multiple regression analysis using the Y-SCS as predictor variables. All analyses were conducted using SPSS®28.

## Results

### Demographic characteristics

Participants were divided into nine regions: American (*n* = 215), Canadian (*n* = 38), Brazilian (*n* = 2), British (*n* = 88), European (*n* = 15), East Asian (*n* = 26), Southeast Asian (*n* = 32), South Asian (*n* = 11), and dual nationality (*n* = 7). See Table 2 for a description of participant characteristics.

### Multilevel regression analysis

I conducted a multilevel regression analysis, nesting participants within regions and allowing the intercept by region to vary randomly. For between-region differences (Level 2), regional means of self-construal were calculated by averaging the respective S-SCS of participants within each region. For each participants' within-region variance (Level 1), regional mean-centred S-SCS was calculated by subtracting the regional means of interdependent and independent self-construals from each participants' S-SCS scores respectively. To minimise Type I error (false positive) since multiple

**Table 2. Participant characteristics across regions.**

| | American | Canadian | Brazilian | British | European | East Asian | Southeast Asian | South Asian | Dual Nationality |
|---|---|---|---|---|---|---|---|---|---|
| | (n=215) | (n=38) | (n=2) | (n=88) | (n=15) | (n=26) | (n=32) | (n=11) | (n=7) |
| | M (SD) | M (SD) | M (SD) | M (SD) | M (SD) | M (SD) | M (SD) | M (SD) | M (SD) |
| Age (Years) | 38.66 (13.07) | 40.74 (12.38) | 26.50 (0.71) | 31.65 (12.07) | 32.08 (10.87) | 28.00 (7.59) | 29.15 (7.60) | 38.70 (11.57) | 28.67 (13.02) |
| Gender, n (%) | | | | | | | | | |
| Transgender | 0 (0.0) | 0 (0.0) | 0 (0.0) | 0 (0.0) | 0 (0.0) | 1 (3.8) | 0 (0.0) | 0 (0.0) | 0 (0.0) |
| Non-binary | 4 (1.9) | 0 (0.0) | 0 (0.0) | 4 (4.5) | 0 (0.0) | 0 (0.0) | 0 (0.0) | 0 (0.0) | 0 (0.0) |
| Female | 105 (48.8) | 18 (47.4) | 1 (50.0) | 32 (36.4) | 7 (46.7) | 22 (84.6) | 17 (53.1) | 7 (63.6) | 2 (28.6) |
| Male | 104 (48.4) | 20 (52.6) | 1 (50.0) | 52 (59.1) | 8 (53.3) | 2 (7.7) | 15 (46.9) | 4 (36.4) | 5 (71.4) |
| Prefer not to say | 2 (0.9) | 0 (0.0) | 0 (0.0) | 0 (0.0) | 0 (0.0) | 0 (0.0) | 0 (0.0) | 0 (0.0) | 0 (0.0) |
| Disability, n (%) | | | | | | | | | |
| Yes | 35 (16.3) | 11 (28.9) | 0 (0.0) | 4 (4.5) | 1 (6.7) | 2 (7.7) | 1 (3.1) | 0 (0.0) | 6 (85.7) |
| No | 173 (80.5) | 26 (68.4) | 2 (100) | 78 (88.6) | 14 (93.3) | 24 (92.3) | 30 (93.8) | 11 (100.0) | 0 (0.0) |
| Prefer not to say | 3 (1.4) | 0 (0.0) | 0 (0.0) | 2 (2.3) | 0 (0.0) | 0 (0.0) | 0 (0.0) | 0 (0.0) | 0 (0.0) |
| Prefer to self-describe | 0 (0.0) | 0 (0.0) | 0 (0.0) | 2 (2.3) | 0 (0.0) | 0 (0.0) | 1 (3.1) | 0 (0.0) | 1 (14.3) |
| Education Level | 4.99 (0.71) | 5.00 (0.82) | 5.50 (0.71) | 4.87 (0.65) | 5.15 (0.90) | 5.26 (0.56) | 4.85 (0.82) | 5.50 (0.53) | 5.00 (0.89) |
| Musical Identity [a] | 2.55 (1.34) | 2.21 (1.07) | 2.50 (0.71) | 2.64 (1.34) | 2.23 (1.17) | 3.42 (1.84) | 2.44 (1.19) | 1.90 (0.57) | 4.17 (1.17) |
| Musical Training (Years) | 4.01 (5.31) | 3.76 (6.09) | 1.00 (1.41) | 4.29 (6.28) | 2.85 (5.97) | 7.11 (8.96) | 3.11 (4.84) | 0.70 (0.95) | 11.5 (6.16) |
| Self-Construal | | | | | | | | | |
| Interdependent | 4.99 (0.91) | 4.64 (0.75) | 4.84 (0.33) | 4.53 (0.72) | 4.37 (0.69) | 4.77 (0.59) | 4.96 (0.71) | 5.28 (0.55) | 4.58 (0.59) |
| Independent | 5.25 (0.85) | 5.06 (0.78) | 5.87 (0.66) | 5.01 (0.69) | 5.02 (0.86) | 4.72 (0.57) | 4.96 (0.73) | 5.15 (0.95) | 4.83 (0.55) |

[a]Musical identity was obtained using the OMSI [60] musician rank item. Musical identity was used because it was reported to be the best single-item measure that represents musical sophistication and musicality [61].

hypothesis tests were conducted on the same dataset for each BMRQ subtype, I applied a Bonferroni correction (.05/ 5) for an adjusted $\alpha$ of.01 [69].

### Role of S-SCS in musical reward

**Social reward.** Results showed that both regional mean-centred interdependent self-construal ($b=1.14$, $SE=0.17$, $p<.001$) and regional mean-centred independent self-construal ($b=1.23$, $SE=0.17$, $p<.001$) were positively associated with social reward. No significant associations were observed for the regional means of interdependent and independent self-construals.

**Musical seeking.** Results showed that only regional mean-centred independent self-construal ($b=1.11$, $SE=0.17$, $p<.001$) was positively associated with musical seeking reward. No significant associations were observed for the regional means of interdependent and independent self-construals.

**Emotion evocation.** Results showed that both regional mean-centred interdependent self-construal ($b=0.68$, $SE=0.16$, $p<.001$) and regional mean-centred independent self-construal ($b=0.44$, $SE=0.16$, $p=.008$) were positively associated with emotion evocation reward. No significant associations were observed for the regional means of interdependent and independent self-construals.

**Mood regulation.** Results showed that only regional mean-centred independent self-construal ($b=0.52$, $SE=0.14$, $p<.001$) was positively associated with mood regulation reward. No significant associations were observed for the regional means of interdependent and independent self-construals.

**Sensory-motor.** Results showed that only regional mean-centred independent self-construal ($b=0.78$, $SE=0.20$, $p<.001$) was positively associated with sensory-motor reward. No significant associations were observed for the regional means of interdependent and independent self-construals.

Taken together, the results showed that only within-region variation of interdependent and independent self-construals, not between-region variation of interdependence and independence, were positively associated with musical reward subtypes (see Table 3).

## Multiple regression analysis

Since both interdependent and independent self-construals were positively associated with several musical reward subtypes (i.e., social reward and emotion evocation), I conducted a multiple regression analysis using the eight dimensions of Y-SCS as an extension and refinement of the two-factor S-SCS model. As the previous analyses found no significant regional differences, all participants were collapsed into a single group for this analysis. To minimise the risk of Type I error (false positive) arising from multiple hypothesis tests conducted on the same dataset for each BMRQ subtype, a Bonferroni correction was applied (.05/ 5), resulting in an adjusted $\alpha$ of .01.

A post hoc power analysis conducted using G*Power 3.1 for the overall $F$ test of a multiple regression ($\alpha=.01$) with $N=435$ and eight predictors indicated that the sample was more than adequately powered ($1-\beta\approx.999$) to detect an effect size of $f^2=.15$. This estimate is consistent with the earlier multilevel regression analyses, which yielded marginal $R^2$ values ranging from .06 to .23, indicating small-to-moderate effect sizes.

## Role of Y-SCS in musical reward

**Social reward.** Results indicated that the model was significant, $R^2=.09$, adjusted $R^2=.07$, $F(8, 426) = 5.22$, $p<.001$. *Self-containment vs. connectedness to others* ($\beta=-.24$, $p=.000$, $sr^2=-.207$) and *self-reliance vs. dependence on others* ($\beta=-.15$, $p=.007$, $sr^2=-.125$) remained significant predictors of social reward.

**Musical seeking.** Results indicated that the model was significant, $R^2=.10$, adjusted $R^2=.09$, $F(8, 426) = 6.02$, $p<.001$. *Difference vs. similar to others* ($\beta=.15$, $p=.005$, $sr^2=.130$) and *self-containment vs. connectedness to others* ($\beta=-.24$, $p=.000$, $sr^2=-.213$) remained significant predictors of musical seeking reward.

**Emotion evocation.** Results indicated that the model was significant, $R^2=.07$, adjusted $R^2=.05$, $F(8, 426) = 3.68$, $p<.001$. *Difference vs. similar to others* ($\beta=.16$, $p=.004$, $sr^2=.135$) remained a significant predictor of emotion evocation reward.

**Mood regulation.** Results indicated that the model was significant, $R^2=.09$, adjusted $R^2=.08$, $F(8, 426) = 5.40$, $p<.001$. *Self-containment vs. connectedness to others* ($\beta=-.27$, $p=.000$, $sr^2=-.236$) remained a significant predictor of mood regulation reward.

**Sensory-motor.** Results indicated that the model was significant, $R^2=.10$, adjusted $R^2=.08$, $F(8, 426) = 5.95$, $p<.001$. *Self-containment vs. connectedness to others* ($\beta=-.26$, $p=.000$, $sr^2=-.229$) remained a significant predictor of sensory-motor reward. See Table 4 for a summary of the multiple regression analysis results, situating each self-construal dimension vis-à-vis the different musical reward subtypes. Refer to S2 Appendix for the full multiple regression analysis results.

In summary, the results indicate that different self-construal profiles were associated with various musical reward subtypes. Specifically, *social reward* was significantly predicted by the interdependent poles of *connectedness to others* and *dependence on others*. *Musical seeking* was significantly predicted by both the independent pole of *difference* and the interdependent pole of *connectedness to others*. *Emotion evocation* was significantly predicted by the independent pole of

**Table 3. multilevel regression analysis results for musical reward using S-SCS as predictors.**

| Predictors | Outcome Variable | | | |
|---|---|---|---|---|
| | *b* | *SE* | *t* | *p* |
| | Social Reward | | | |
| Interdependent$_{RegionalMean}$ | 1.00 | 0.77 | 1.29 | .271 |
| **Interdependent$_{RegionalMeanCentred}$** | **1.23** | **0.17** | **7.40** | **<.001** |
| Independent$_{RegionalMean}$ | −1.00 | 1.03 | −0.97 | .380 |
| **Independent$_{RegionalMeanCentred}$** | **1.14** | **0.17** | **6.62** | **<.001** |
| | Musical Seeking | | | |
| Interdependent$_{RegionalMean}$ | 0.77 | 0.82 | 0.94 | .396 |
| Interdependent$_{RegionalMeanCentred}$ | 0.17 | 0.16 | 1.03 | .304 |
| Independent$_{RegionalMean}$ | −1.26 | 1.08 | −1.16 | .295 |
| **Independent$_{RegionalMeanCentred}$** | **1.11** | **0.17** | **6.69** | **<.001** |
| | Emotion Evocation | | | |
| Interdependent$_{RegionalMean}$ | −0.85 | 0.69 | −1.23 | .219 |
| **Interdependent$_{RegionalMeanCentred}$** | **0.68** | **0.16** | **4.29** | **<.001** |
| Independent$_{RegionalMean}$ | 0.65 | 0.93 | 0.70 | .486 |
| **Independent$_{RegionalMeanCentred}$** | **0.44** | **0.16** | **2.67** | **.008** |
| | Mood Regulation | | | |
| Interdependent$_{RegionalMean}$ | 0.54 | 0.58 | 0.94 | .348 |
| Interdependent$_{RegionalMeanCentred}$ | 0.32 | 0.13 | 2.37 | .018 |
| Independent$_{RegionalMean}$ | −1.11 | 0.78 | −1.42 | .158 |
| **Independent$_{RegionalMeanCentred}$** | **0.52** | **0.14** | **3.76** | **<.001** |
| | Sensory-Motor | | | |
| Interdependent$_{RegionalMean}$ | 1.14 | 1.01 | 1.13 | .301 |
| Interdependent$_{RegionalMeanCentred}$ | 0.45 | 0.19 | 2.32 | .021 |
| Independent$_{RegionalMean}$ | 0.45 | 1.34 | 0.34 | .743 |
| **Independent$_{RegionalMeanCentred}$** | **0.78** | **0.20** | **3.92** | **<.001** |

*Note.* $N = 435$. Regional-means of interdependent and independent self-construals were calculated by averaging the respective self-construals of participants within each region. This was to reflect between-region differences in self-construal (Level 2). Regional mean-centred self-construals were calculated by subtracting the regional-means of interdependent and independent self-construals from each participants' interdependent and independent self-construals respectively. This was to reflect each participants' within-region variance in self-construal (Level 1). Bold values indicate statistical significance (using an $\alpha$ of .01).

*difference*, whereas *mood regulation* and *sensory-motor* subtypes were significantly predicted by the interdependent pole of *connectedness to others*.

## Discussion

The primary aim of this exploratory study was to investigate whether self-construal, both between and within cultures, was associated with different musical rewards when listening to favourite music. Results from the multilevel regression analyses showed that between-region variations in interdependence and independence were not associated with musical reward, suggesting that musical reward and its subtypes are comparable between geographical regions. I recognise, however, that this lack of difference may be partly due to how participants were grouped. Given the heterogeneity of the sample, I applied the cultural distance hypothesis to group participants, enabling the aggregation of data across regions

**Table 4. Summary of multiple regression analysis results for musical reward using Y-SCS as predictors.**

| Predictor | Musical Reward Subtypes | | | | |
|---|---|---|---|---|---|
| | Social Reward | Musical Seeking | Emotion Evocation | Mood Regulation | Sensory-Motor |
| Difference vs. similar to others | 0.07 | **0.15**** | **0.16**** | 0.09 | 0.10 |
| Self-containment vs. connectedness to others | **−0.24***** | **−0.24***** | −0.10 | **−0.27***** | **−0.26***** |
| Self-direction vs. reception to influence | −0.01 | −0.03 | 0.10 | 0.09 | 0.13 |
| Self-expression vs. harmony | 0.05 | 0.10 | 0.07 | 0.08 | 0.05 |
| Consistency vs. variability | 0.13 | 0.11 | −0.12 | −0.01 | 0.08 |
| Decontextualized vs. contextualized self | −0.05 | −0.10 | −0.03 | 0.01 | 0.01 |
| Self-reliance vs. dependence on others | **−0.15**** | −0.01 | −0.07 | 0.07 | −0.05 |
| Self-interest vs. commitment to others | 0.07 | 0.13 | −0.11 | 0.03 | −0.01 |

*Note.* This table presents the standardized $\beta$ coefficients for all eight self-construal dimensions across the musical reward subtypes. Positive beta coefficients indicate a positive association with the independent pole of the Y-SCS factor whereas negative beta coefficients indicate a positive association with the interdependent pole of the Y-SCS factor. Bold values indicate statistical significance (using an $\alpha$ of .01).

** $p < .01$, *** $p < .001$.

for greater statistical power [68]. For one, the uneven distribution of participants between regions may have diminished statistical power to observe differences between regions. Alternatively, these findings align with previous research which found that assuming country proxy for collectivism-individualism may be insufficient for detecting cultural variations [70,71]. Nonetheless, it should not necessarily be concluded that the determinants of musical reward associated with favourite music are pancultural. Cultural determinants exist not only in the sociocultural contexts in which individuals reside, but also within individuals through psychological constructs [72].

Results from the multilevel regression analyses further revealed that within-region variation of interdependent and independent self-construals were associated with musical rewards. My first hypothesis was partially supported insofar as interdependent self-construal was positively associated with social reward. This finding aligns with prior research showing that members of collectivistic societies tend to use music for social bonding [31–33]. Importantly, these results extend previous between-country studies by highlighting that interdependent self-construal, found within regions, was positively linked to social reward. In other words, interdependent self-construal may explain within-cultural variation in social reward experienced with favourite music. The results further revealed that within-region variation of independent self-construal was also positively associated with social reward. I speculate that listening to one's favourite music is not only a self-directed personal activity – which explains the observed association with independent self-construal – but also helps people connect with others. This supports the idea that music is universally important for social cohesion [6,73–75].

Contrary to my first hypothesis, this study found that interdependent self-construal was also positively associated with emotion evocation reward. This result was unexpected but not entirely surprising. Research in cultural psychology has found an association between interdependent self-construal and socially engaging emotions – affective states that foster social bonding [76–80]. In addition, recent work in the psychology of music has shown that some individuals describe music-listening experiences as emotional encounters involving a felt sense of connection [75]. Consequently, it is possible that people with a dominant interdependent self-construal may listen to their favourite music to evoke socially engaging emotions so as to feel more connected with both real and imagined others. It would be worth testing this hypothesis in future.

My second hypothesis was partially supported. First, the findings indicated that independent self-construal was positively associated with musical seeking reward. This suggests that the rewards derived from seeking information about music are driven by personal motivation and interest. This makes sense, as engaging in musical exploration is effortful, requiring self-motivation and curiosity, which aligns with the characteristics of an independent self-construal. This positive

association highlights the role of independent self-construal in musical seeking reward experienced with favourite music. Second, the findings showed that independent self-construal was positively associated with mood regulation reward. This is consistent with previous research indicating that members of individualistic societies often use music for emotion regulation [31–33]. These results extend this body of work by demonstrating that mood regulation reward derived from favourite music is salient for individuals with a dominant independent self-construal. Third, the findings revealed that independent self-construal was positively associated with sensory-motor reward. Research suggests that dancing to music stimulates endorphin release which enhances mood [74]. In other words, moving to music may reinforce a sense individuality and personal expression.

As described earlier, the multilevel regression analyses showed that both interdependent and independent self-construals were positively associated with several musical reward subtypes (i.e., social reward and emotion evocation). While the two-dimensional model of self-construal offers theoretical parsimony, it may obscure specific psychological mechanisms of self-construal underlying inter- and intra-cultural differences [70]. Consequently, an exploratory approach was adopted using the eight-dimensional model of self-construal [58] to further delineate aspects of self-construal related to musical reward.

Consistent with the earlier analyses involving the S-SCS, results from the multiple regression analyses using the Y-SCS partially supported my hypotheses. Specifically, the interdependent poles of *connectedness to others* and *dependence on others* were positively associated with social reward, while the independent pole of *difference* was associated with musical seeking and emotion evocation rewards.

However, the multiple regression analyses also revealed some differences. Notably, the interdependent pole of *connectedness to others* was positively associated with musical seeking, mood regulation, and sensory-motor rewards – findings that contrast with the earlier results showing associations with independent self-construal only. This discrepancy suggests that the S-SCS may lack the sensitivity and precision to capture nuanced aspects of interdependent and independent self-construals in relation to musical reward. Alternatively, it is possible that the analytical approach, which did not account for cultural context in the multiple regression analyses, contributed to these differences in findings.

Nevertheless, one finding stood out: the interdependent pole of *connectedness to others* was positively associated with most musical reward subtypes (except emotion evocation). This provides further evidence for the notion that music serves as a universal means of fostering social bonding [6,73,74]. Ultimately, these findings should be regarded as exploratory and interpreted with caution.

One strength of the present study is its inclusion of specific cultural factors – namely, self-construal – in examining the cultural determinants of musical rewards. The use of the two-dimensional model of self-construal provides initial evidence for its role in shaping the types of musical rewards experienced with favourite music across and within cultural contexts. Moreover, the incorporation of the eight-dimensional model offers a more fine-grained understanding of how distinct aspects of self-construal relate to particular subtypes of musical reward. Together, these findings underscore the importance of integrating both cultural context and individual-level cultural factors in advancing a more nuanced understanding of the role of culture in music experiences. As the first cross-cultural study on musical reward, this research establishes a foundation for future work to further explore the determinants of musical reward both across and within cultures.

Beyond research implications, these findings hold practical relevance for music therapists and arts and health practitioners working with culturally diverse populations. Taking a cultural approach to well-being [81] requires recognising that individuals from different cultural backgrounds may hold unique self-construal profiles, which in turn shape their health priorities [82]. By understanding how self-construal relates to different subtypes of musical reward, practitioners can design music-based interventions that are culturally appropriate, sensitive, and responsive to their clients' diverse needs, thereby promoting health and well-being more effectively.

Four potential limitations of the current study should be considered. First, this study examined musical rewards within the context of the individual's favourite music, which varies widely among listeners and may affect the type of rewards

experienced. For instance, participants may have selected contemporary music that is culturally similar due to the homogenising effects of globalisation [83], potentially contributing to the lack of observed differences in musical rewards between cultures. Also, participants were asked to reflect on their 'all-time' favourite music, which may have led some to select music that has remained significant and enjoyable to people over the years. While this is possible, I argue that individuals with a dominant interdependent self-construal are more likely to choose such music, as their preferences may be strongly influenced by others. As discussed earlier, favourite music was used in the present study to provide a snapshot of the relationship between self-construal, found across and within cultures, and musical reward. In future, researchers could examine how specific elements of music are related to self-construal and musical reward.

Second, a cross-sectional survey was administered and correlational tests applied to the data, so no conclusions can be drawn as to whether self-construal causes musical reward or vice versa. Rather, I speculate that self-construal and experiences of listening to music have a mutual influence on each other, as do the self and cultural factors [9,12]. Third, there was a risk of self-selection bias such that only people already invested in music participated in the study, leading to an overestimation of the strength of the relationship between self-construal and musical reward. For example, anecdotal evidence from one participant wrote, "I had to write and let you know how much I enjoyed this survey. I am an extreme music lover so I really got into this one, thanks!" In addition, this study partially relied on MTurk participants, which potentially limits the generalizability of the findings [84]. Fourth, the heterogenous sample was divided into various regions, meaning that multiple nationalities and countries were included in each group. As previously noted, I acknowledge that culture is not neatly bounded within these sociodemographic categories. Future research should address this limitation by recruiting participants from specific countries to allow for a more focused exploration of how self-construal influences musical reward.

The results of this study are dependent on the way I have chosen to conceptualise and operationalise cultural factors and musical experiences. Other cultural factors that were not explored in this study, such as religious heritage [85,86] and cultural worldviews [71,87,88], may influence musical rewards. Nevertheless, these findings suggest that cultural context is a necessary but insufficient aspect of cultural variation, underscoring the importance of combining theories of culture with methodological innovations that do justice to a nuanced understanding of how cultural specificities influence music experience [7–9]. In future, research could explore the intersection of cultural imperatives and musical cultures. For example, in certain cultures (e.g., Brazilian and African cultures), dancing to music may be considered a way of expressing one's connectedness with others [89,90], whereas in other cultures (e.g., Chinese culture), dancing to music may be perceived as more self-enhancing [21]. Future researchers could also examine the content of the music and investigate other musical activities such as composing and playing instruments or singing. Previous studies have found that lyrics of pop songs contained collectivistic and individualistic themes [91–93], which may give rise to different self-construals and influence the types of musical reward experienced.

In conclusion, this study provides first evidence of the role of self-construal in shaping musical rewards associated with one's favourite music across and within cultural contexts. Using the two-dimensional model of self-construal, both interdependent and independent self-construals were found to be positively associated with emotion evocation and social reward, while independent self-construal was positively associated with musical seeking, mood regulation, and sensory-motor subtypes. When applying the eight-dimensional model, distinct self-construal profiles emerged in relation to different musical reward subtypes, with the interdependent pole of *connectedness to others* positively associated with most subtypes except for emotion evocation reward. In essence, our sense of self, particularly the extent to which we perceive ourselves as interconnected or independent, affects the kind of pleasure we derive from listening to our favourite music. As the first study of its kind, these findings lay important groundwork for future cross-cultural research on musical reward. By adopting a novel approach to cross-cultural investigations in music psychology, this study underscores the significance of self-construal as an individual-level explanation for culturally based differences in affective experiences with music, thereby contributing to a more nuanced understanding of cultural diversity in music psychology.

## Supporting information

**S1 Appendix. Participant characteristics.**
(DOCX)

**S2 Appendix. Full multiple regression analysis results.**
(DOCX)

## Acknowledgments

I would like to thank Nikki Dibben and Renee Timmers for their feedback on earlier versions of this article.

## Author contributions

**Conceptualization:** Jonathan Tang.

**Data curation:** Jonathan Tang.

**Formal analysis:** Jonathan Tang.

**Funding acquisition:** Jonathan Tang.

**Investigation:** Jonathan Tang.

**Methodology:** Jonathan Tang.

**Project administration:** Jonathan Tang.

**Resources:** Jonathan Tang.

**Supervision:** Jonathan Tang.

**Validation:** Jonathan Tang.

**Visualization:** Jonathan Tang.

**Writing – original draft:** Jonathan Tang.

**Writing – review & editing:** Jonathan Tang.

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
