## [Editor Report · Decision Letter 0]

2 Jun 2025

Dear Dr. Tang,

Thank you for submitting your manuscript to PLOS ONE. After careful consideration, we feel that it has merit but does not fully meet PLOS ONE’s publication criteria as it currently stands. Therefore, we invite you to submit a revised version of the manuscript that addresses the points raised by the academic editor befor sending it out for review. Please find the commnets below.

We look forward to receiving your revised manuscript.

Kind regards,

Seung-Goo Kim, Ph.D.

Academic Editor

PLOS ONE

“This work was supported by the Arts & Humanities Research Council (grant number AH/R012733/1) through the White Rose College of the Arts & Humanities.”

“I would like to thank Nikki Dibben and Renee Timmers for their feedback on earlier versions of this article. This work was supported by the Arts & Humanities Research Council (grant number AH/R012733/1) through the White Rose College of the Arts & Humanities.”

“This work was supported by the Arts & Humanities Research Council (grant number AH/R012733/1) through the White Rose College of the Arts & Humanities.”

Additional Editor Comments:

Both MANCOVA (page 11) and multiple regressions (page 12) need to be controlled for the inlfated Type-I error. PLOS ONE accepts manuscripts not based on the impacts of findgins, but the rigor of methods. Currently, the familiy-wise Type-I error appears largely inflated.

---

## [Author Response · Author response to Decision Letter 1]

4 Jul 2025

1. I have edited the manuscript to be consistent with the PLOS ONE's style requirements.

2. The funders had no role. I have amended this in my cover letter.

3. I have removed the funding information from my acknowledgements. I have also updated my cover letter accordingly.

4. Noted. My entire data will be made freely accessible if my manuscript is accepted for publication. I have updated my data availability statement accordingly on the submission form.

5. I have included the captions for my Supporting Information files at the end of the manuscript. I have also updated the in-text citations to match accordingly.

Additional editor comments:

For MANCOVA, I believe there is a higher risk of Type-II error (rather than Type-I error) given that I have included several covariates. So, I have left it as is.

Thank you for the suggestion for controlling for Type-I error for the multiple linear regressions. I have applied a Bonferroni correction and edited my results/discussion sections accordingly.

---

## [Decision Letter · Decision Letter 1]

27 Aug 2025

Dear Dr. Tang,

Thank you for submitting your manuscript to PLOS ONE. After careful consideration, we feel that it has merit but does not fully meet PLOS ONE’s publication criteria as it currently stands. Therefore, we invite you to submit a revised version of the manuscript that addresses the points raised during the review process.

We look forward to receiving your revised manuscript.

Kind regards,

Seung-Goo Kim, Ph.D.

Academic Editor

PLOS ONE

Journal Requirements:

**Additional Editor Comments:**

In particular, the current study partly relies on MTurk participants, whose validity has been questioned and for whom many pitfalls have been discussed. See https://doi.org/10.1371/journal.pone.0057410 and https://doi.org/10.1146/annurev-clinpsy-021815-093623 for reference. Please discuss the potential limitations of this sample when interpreting the present results.

Reviewers' comments:

Reviewer's Responses to Questions

**Comments to the Author**

Reviewer #1: (No Response)

2. Is the manuscript technically sound, and do the data support the conclusions?

Reviewer #1: Partly

3. Has the statistical analysis been performed appropriately and rigorously?

Reviewer #1: No

4. Have the authors made all data underlying the findings in their manuscript fully available?

Reviewer #1: No

5. Is the manuscript presented in an intelligible fashion and written in standard English?

Reviewer #1: Yes

Reviewer #1: 1. General Assessment

The manuscript addresses a timely and potentially valuable topic related to personality traits, self-construals, and collectivism-individualism distinctions within a cross-cultural psychological framework. The authors’ attempt to explore subtle dynamics between personality dimensions (as measured through the BFI) and cultural self-views (independent vs. interdependent self-construals) reflects a clear alignment with ongoing theoretical efforts in cultural psychology.

The research is novel in its focus on linking self-construal types with Big Five dimensions within a specific cultural context, which is under-represented in the literature. The study is based on a reasonably large sample (N=435), and the statistical analyses are generally appropriate.

However, there are substantial conceptual and methodological limitations that require thorough addressing before the manuscript can be considered for publication.

2. Major Concerns

2.1. Grouping Procedure and Conceptual Validity

A critical issue undermining the paper’s internal validity lies in the arbitrary dichotomization of participants into two groups – “collectivists” (n=73) and “individualists” (n=361). Aside from the numerical inconsistency (73 + 361 = 434, not 435 – this should be explicitly corrected), the categorization threshold appears both statistically and conceptually fragile.

Someone scoring 49 is considered “collectivist”, while someone scoring 51 is “individualist”, although these individuals may be typologically indistinguishable. No buffer zone or standard deviation criterion was applied to ensure clear group separation, which significantly weakens the interpretation of group differences.

I strongly recommend using a buffer zone (e.g., excluding cases close to the median, or applying a 0.5 SD band around the cut-off point) to enhance group distinctiveness – even at the cost of reducing subsample sizes. Without this, the derived typologies risk being statistically artificial and theoretically ambiguous, casting doubt over the discussion’s strength.

2.2. Psychometric Weakness of the BFI

The psychometric indicators reported for several BFI scales (e.g., Openness α = .26; Agreeableness α = .48; Conscientiousness α = .51) are unacceptably low for scientific reporting. While the BFI is a widely used tool, its use in the present study raises red flags due to these sub-threshold reliability scores. Even the most liberal acceptance levels for internal consistency require Cronbach’s α > .60 (ideally ≥ .70).

Given that many of the conclusions rest on personality trait comparisons, the low reliability coefficients seriously undermine the robustness of findings. The authors should consider:

• either excluding problematic subscales from analysis,

• or using a validated alternative instrument with stronger psychometric support for their target population.

2.3. Theoretical Framing of Self-Construals

The manuscript relies on a binary construal of independent vs. interdependent self, with strong associations to the individualism–collectivism dimension. However, recent advances (e.g., Vignoles et al., 2016) challenge this simplistic East–West dichotomy and instead advocate for multidimensional models of selfhood. These include domains such as:

• self-direction vs. heteronomy,

• differentiation vs. similarity,

• self-containment vs. connectedness.

I recommend the authors critically engage with such multifaceted frameworks of selfhood and reflect on their potential implications for interpretation of results. This would substantially improve the theoretical sophistication of the manuscript.

2.4. Lack of Power Analysis and Uneven Group Sizes

Another important omission is the lack of a prior statistical power analysis, which would have been critical given the asymmetric group sizes (n=73 vs. n=361). The rationale for this division is already questionable (see 2.1), but even assuming such a dichotomy is acceptable, the authors should have:

• justified whether the smaller group size (n=73) provides adequate power to detect medium or small effects;

• reported effect size sensitivity and alpha thresholds adopted in the study;

• possibly explored balanced or bootstrapped re-sampling techniques to control for sample-size-induced bias.

Given that all conclusions hinge on this between-group comparison, the current design may lead to both inflated Type I errors (due to multiple testing) and Type II errors (in the underpowered subgroup). The manuscript would greatly benefit from including:

• a post hoc power analysis, at minimum, and preferably;

• a justification for group sizes based on a priori power estimations aligned with expected effect sizes (e.g., based on literature or pilot studies).

3. Additional Issues and Clarifications

• Statistical Validity & Type I Error:

Regarding the editor's observation on Type I error inflation, the current analysis would indeed benefit from correction for multiple comparisons. Bonferroni correction, while conservative, is an acceptable first step to mitigate this issue. However, given the large number of comparisons, the authors might also consider:

o Holm-Bonferroni (a sequentially rejective procedure),

o or False Discovery Rate (FDR) methods (e.g., Benjamini–Hochberg), which preserve power while reducing false positives.

Regardless of the method chosen, this correction must be transparently reported and integrated into result interpretation.

• Clarification needed on the rationale for using two-sample t-tests instead of more robust multivariate techniques (e.g., MANOVA), given the multiple dependent variables and group comparisons.

• Literature is overly reliant on early 2000s sources. Authors are encouraged to include and engage with more recent work, such as:

Beugelsdijk, S., & Welzel, C. (2018). Dimensions and dynamics of national culture: Synthesizing Hofstede with Inglehart. Journal of Cross-Cultural Psychology, 49(10), 1469–1505. https://doi.org/10.1177/0022022118798505 – discussing dimensions beyond I-C.

Church, A. T. (2016). Personality traits across cultures. Current Opinion in Psychology, 8, 22–30. https://doi.org/10.1016/j.copsyc.2015.09.014 – examining Big Five validity across cultures.

Vignoles, V. L., Owe, E., Becker, M., Smith, P. B., Easterbrook, M. J., Brown, R., ... & Bond, M. H. (2016).

Beyond the ‘east–west’ dichotomy: Global variation in cultural models of selfhood. Journal of Experimental Psychology: General, 145(8), 966–1000. https://doi.org/10.1037/xge0000175 (already mentioned).

4. Minor Issues

• Table formatting: several tables are overly dense and require clearer presentation (e.g., exact p-values, standardized effect sizes).

• Language: minor grammatical issues persist – a native-level proofreading would benefit clarity and flow.

• Reference formatting: Reference formatting should be checked for full compliance with PLOS ONE’s style guide. Although journal names are not italicized in this format, some inconsistencies are noticeable (e.g., missing volume numbers or inconsistent punctuation). Please ensure that all entries follow the required journal template.

5. Recommendation

While the study has potential and touches upon a meaningful and underexplored intersection between self-construals and personality, the issues above are non-trivial. They require both methodological and theoretical reconsideration. In this context, I suggest a thorough revision of the manuscript, addressing the following key points:

• clear redefinition of group assignment criteria;

• addressing low reliability metrics;

• integrating updated theoretical perspectives;

• correcting for multiple comparisons;

• enhancing statistical reporting transparency.

**Do you want your identity to be public for this peer review?** For information about this choice, including consent withdrawal, please see our Privacy Policy

Reviewer #1: **Yes: ** Alin Gavreliuc

---

## [Author Response · Author response to Decision Letter 2]

21 Oct 2025

Thank you for taking the time to review my article and for the opportunity to edit my paper. I have attached my responses to the reviewer and editors comments.

---

## [Decision Letter · Decision Letter 2]

1 Dec 2025

Dear Dr. Tang,

Thank you for submitting your manuscript to PLOS ONE. After careful consideration, we feel that it has merit but does not fully meet PLOS ONE’s publication criteria as it currently stands. Therefore, we invite you to submit a revised version of the manuscript that addresses the points raised during the review process.

In particular, Reviewer 1 noted that there appears to be an inconsistency in the reported timeline of the research, which I hope the author can clarify.

We look forward to receiving your revised manuscript.

Kind regards,

Seung-Goo Kim, Ph.D.

Academic Editor

PLOS ONE

Journal Requirements:

Reviewers' comments:

Reviewer's Responses to Questions

**Comments to the Author**

Reviewer #1: All comments have been addressed

Reviewer #2: (No Response)

2. Is the manuscript technically sound, and do the data support the conclusions?

Reviewer #1: Partly

Reviewer #2: Yes

3. Has the statistical analysis been performed appropriately and rigorously?

Reviewer #1: Yes

Reviewer #2: Yes

4. Have the authors made all data underlying the findings in their manuscript fully available?

Reviewer #1: Yes

Reviewer #2: Yes

5. Is the manuscript presented in an intelligible fashion and written in standard English?

Reviewer #1: Yes

Reviewer #2: Yes

Reviewer #1: General Assessment

I appreciate the authors’ clear effort to substantially revise and refine this manuscript in light of prior feedback. The new version demonstrates both conceptual and methodological growth. In particular, the inclusion of a multifaceted self-construal measure (based on Vignoles et al., 2016, and the Y-SCS for the Chinese sample) marks a significant theoretical and empirical advancement. This addition allows for a far more nuanced understanding of self-related variation in musical reward and provides a welcome alternative to the earlier binary conceptualization (independent vs. interdependent). The revised analyses are pertinent, carefully interpreted, and better aligned with the study’s stated goals.

Overall, the manuscript now presents a richer and more theoretically coherent contribution to the emerging intersection between cultural psychology and music cognition. The authors’ responses to earlier critiques were thoughtful and, in most cases, convincingly addressed.

Major Issue for Clarification

The only serious reservation I still hold concerns the temporal and procedural sequence related to the adoption of the new self-construal instrument.

According to the Methods section (p. 17), the authors acknowledge that they obtained the multifaceted self-construal scale from Dr. Viv Vignoles after their initial correspondence (Aug 22, 2022) and receipt of the materials on Nov 15, 2022. However, data collection reportedly began on Nov 3, 2022—almost two weeks earlier.

This raises an important question of study design integrity:

- Was data collection initiated before the complete set of study instruments and design decisions were finalized?

- If so, how was the new self-construal measure incorporated—was it added mid-study, or were some participants excluded and recollected?

- If these data were indeed available at the time of the first submission, why were results from this more appropriate instrument not reported earlier?

Clarifying this procedural timeline is crucial, because it directly affects the transparency and replicability of the research. The issue does not necessarily invalidate the findings, but it requires an explicit methodological statement in the manuscript (e.g., under “Limitations” or “Procedure”) to ensure that readers and editors understand how the final dataset corresponds to the revised study design.

Minor Comments and Suggestions

1. Methodological transparency:

Please indicate explicitly (a) whether the full sample (N = 435) completed the multifaceted self-construal measure, and (b) whether any earlier data were discarded or replaced when adopting the new instrument.

2. Theoretical framing:

The integration of the eight-dimensional self model could be strengthened by explicitly situating each dimension vis-à-vis the observed musical reward subtypes—for instance, noting how “self-direction vs. heteronomy” aligns with the musical seeking and mood regulation factors.

3. Statistical reporting:

Although regression results are clearly reported, the presentation could be further improved by including a single summary table (perhaps in the Supplementary Material) that lists standardized β coefficients for all eight self-construal dimensions across the musical reward subtypes, for easier comparison.

Reviewer #2: Thank you for the opportunity to review the article entitled ‘"I feel good... I knew that I would...": The role of self in musical reward across cultures’, submitted to the PLOS ONE.

This study explores cultural differences in various types of musical reward experiences. It takes a novel approach by defining culture in terms of self-construal rather than nationality or the individualism–collectivism distinction based on nationality. The findings partially support the hypotheses, revealing different associations between music-related reward types and interdependent versus independent self-construal. Overall, I found the manuscript engaging, well-written, and suitable for publication. The introduction clearly leads into the research question and situates it within the relevant literature. The methods and results are well presented, and the discussion offers nuanced interpretations and thoughtful suggestions for future work. I also reviewed the previous reviewer’s comments and the authors’ revisions. I have only a few minor suggestions that may help strengthen the manuscript.

1. Hypotheses: It was not entirely clear to me why independent self-construal was expected to correlate with specific BMRQ subscales (musical seeking, emotion evocation, mood regulation, and sensory-motor rewards). The rationale for these particular predictions did not seem fully grounded in prior literature. For instance, as you note in the discussion, emotional responses to music can involve both self-focused emotions (e.g., feeling strong, secure, happy) and more social or relational emotions (e.g., awe, transcendence). Similarly, mood regulation through music may include socially oriented processes such as reducing loneliness or engaging in parasocial interaction (e.g., Bannister et al., 2025, Music & Science). I suggest clarifying the theoretical basis for these expectations; or, if more appropriate, addressing this literature in the discussion or future directions.

2. Personality measure (response to reviewer 1 Based on the previous review, I understand that a personality measure and its associated analyses were initially included, but then removed due to low internal consistency. I agree that excluding the scale is appropriate in this case. However, for transparency, it may be helpful to add a brief explanation (even as a footnote) about why the measure was initially included and ultimately omitted.

3. Post-hoc power analysis: Thank you for adding the requested power analysis. However, it would be important to justify the expected effect size.

**Do you want your identity to be public for this peer review?** For information about this choice, including consent withdrawal, please see our Privacy Policy

Reviewer #1: **Yes: ** Alin Gavreliuc

Reviewer #2: No

---

## [Author Response · Author response to Decision Letter 3]

19 Dec 2025

Thank you for taking the time to review my revised manuscript. Regarding the timeline of the research, I can confirm that data collection began after the complete set of study instruments were finalized. I have triple-checked my data records to ensure that this is true. Ethical approval was received on 3 November 2022 with an expected project end date of 7 October 2023 - this was originally reported in the paper. Data collection only began after the study materials were finalized, with the first participant recruited on 27 November 2022 and the last participant on 25 March 2023. I have edited the text to reflect this timeline more accurately on p. 9. I have considered all other feedback and incorporated them where appropriate.

---

## [Editor Report · Decision Letter 3]

23 Dec 2025

"I feel good... I knew that I would...": The role of self in musical reward across cultures

PONE-D-25-25097R3

Dear Dr. Tang,

We’re pleased to inform you that your manuscript has been judged scientifically suitable for publication and will be formally accepted for publication once it meets all outstanding technical requirements.

Kind regards,

Seung-Goo Kim, Ph.D.

Academic Editor

PLOS One

Additional Editor Comments (optional):

I believe the additional points raised by both reviewers have been well addressed.
---

## [Editor Report · Acceptance letter]

PONE-D-25-25097R3

PLOS One

Dear Dr. Tang,

I'm pleased to inform you that your manuscript has been deemed suitable for publication in PLOS One. Congratulations! Your manuscript is now being handed over to our production team.

Kind regards,

on behalf of

Dr. Seung-Goo Kim

Academic Editor

PLOS One